# Olive Mill Wastewater (OMW) Treatment Using Photocatalyst Media

Abeer Al Bawab [1,2,*], Muna Abu-Dalo [3,*], Aya Khalaf [4] and Duaa Abu-Dalo [1]

[1] Chemistry Department, School of Science, The University of Jordan, Amman 11942, Jordan; do.abodalo@yahoo.com
[2] Hamdi Mango Center for Scientific Research, The University of Jordan, Amman 11942, Jordan
[3] Chemistry Department, Faculty of Science and Arts, Jordan University of Science & Technology, Irbid 22110, Jordan
[4] Department of Basic Sciences, Faculty of Arts and Sciences, Al-Ahliyya Amman University, Amman 19328, Jordan; a.khaled@ammanu.edu.jo
[*] Correspondence: drabeer@ju.edu.jo (A.A.B.); maabudalo@just.edu.jo (M.A.-D.)

**Abstract:** A new nanophotocatalysts series of $M_2Zr_2O_7$ (M = Mn, Cu, and Fe) and doped $Fe_2Zr_2O_7$ systems were prepared via sol-gel using the *pechini method*, characterized, and tested in photocatalytic degradation of olive mill wastewater (OMW). The photocatalytic degradation of the prepared materials was evaluated by measuring total phenolic compounds (TPCs) using the *Folin-Ciocalteu method* for variable pH under a commercial LED lamp (45 W). The removal of TPCs was measured at different contact times ranging from 2 h to 6 days. X-ray diffraction (XRD) and transmission electron microscope (TEM) analysis approved the nano size of (5–17 nm) and quasi-spherical morphology of the prepared materials. ICP-OES analysis confirmed the XRD analysis and approved the structure of the prepared materials. Aggregation of the nanomaterials was observed using TEM imaging. Brunauer-Emmett-Teller (BET) analysis measured a 67 $m^2/g$ surface area for $Fe_2Zr_2O_7$. Doping Fe with Mn increased the surface area to 173 $m^2/g$ and increased to 187 $m^2/g$ with a further increase of the Mn dopant. Increasing the Mn dopant concentration increased both surface area and photocatalytic degradation. The highest degradation of TPCs was observed for $Mn_2Zr_2O_7$ around 70% at pH 10 and exposure time up to one day.

**Keywords:** nanophotocatalysts; phenolic compounds; olive mill wastewater (OMW); sol-gel





## 1. Introduction

Water is considered the most essential element for the survival of human life, animals, and plants. Limited water resources and population growth in Jordan increase the need for low-cost, renewable, and non-conventional methods to maximize the available water supply [1]. Treatment of industrial wastewater became one of the alternative approaches to face water scarcity in arid and semi-arid areas, such as Jordan [2]. Olive oil production is considered a major industry in Jordan. According to the Department of Statistics of Jordan (2018), olive farming (12 million trees) covers 24% of Jordan's total arable surface area, and 74% of this area is planted with fruit trees [3]. Olive oil extraction generates olive mill wastewater (OMW), also known as Zebar [4]. This organic waste is one of the most polluting effluents because it contains high levels of phenols, organic compounds, chemical oxygen demand (COD), biological oxygen demand (BOD), microorganisms, nutrients, and toxic compounds [5]. The majority of Jordanian mills discharge their OMW without further treatment due to a lack of knowledge, complexity, and affordability for treatment and/or transport to a landfill site [3]. Many methods have been developed to recover bioactive chemicals and phenolic compounds from olive mill wastewater and recycle treated OMW in the agricultural sector, a potential solution to counteract the water shortage [6]. These methods include adsorption using different low-cost adsorbents [7], biological degradation [8], membrane separation [9], coagulation-flocculation processing [10], filtration [11], and advanced oxidation processes (AOPs) [12].

AOPs involve water treatment at ambient temperature and pressure. These methods are based on the generation of hydroxyl and other radicals as intermediates capable of degrading organic matter [13]. AOPs include chemical processes such as Fenton's reaction [14], Fenton-like reaction [15], and ozonation [16], photochemical processes, such as photolysis of $H_2O_2$ [17] or $O_3$ [18], photo-Fenton reaction [19], and photocatalysis [20], sonochemical processes [21], electrochemical anodic oxidation [22], electro-Fenton (EF) processes [23], and hybrid treatments [24,25].

Gernjak et al. found that adding electron acceptors to titanium dioxide enhanced the degradation properties, although the performance decreased at low pH. This process was accomplished using the photo-Fenton reaction, and the removal of chemical oxygen demand (COD) and phenol index from OMW was 85% and 100%, respectively [26]. An ultrasound/UV/$TiO_2$ system was used by Al-Bsoul et al. to treat OMW with 59% COD removal within 90 min [27]. Iboukhoulef et al. used different system combinations to degrade OMW—ozone, $BiFeO_3$, $O_3/BiFeO_3$, $O_3/H_2O_2$, $O_3/H_2O_2/BiFeO_3$, and $O_3/BiFeO_3/S_2O_8^{2-}$. The combination of $O_3/BiFeO_3/S_2O_8^{2-}$ in alkaline conditions was the most efficient, with 82.9% and 98.0% removal of phenolic compounds and COD, respectively [28].

The advantages of using a photocatalysis method over other advanced oxidation process (AOP) methods include elimination of hazardous chemicals (e.g., ozone and hydrogen peroxide), elimination of waste streams, improved energy efficiency, and flexible light wavelength range activation using doping, which produces a more effective band gap [29]. Many photocatalysts have been used to degrade organic compounds (e.g., phenolic compounds) for wastewater treatment applications. In 1972, titanium dioxide ($TiO_2$) was the first photocatalyst used for water splitting into hydrogen and oxygen by Fujishima and Honda [30]. In 1991, Bahnemann et al. developed a novel method for water detoxification by using $TiO_2$ suspension with promising groundwater and wastewater treatment applications [31]. Thiruvenkatachari et al. reported a higher photocatalytic efficiency and stability for $TiO_2$ in aqueous media compared with $\alpha$-$Fe_2O_3$, $ZrO_2$, CdS, $WO_3$, and $SnO_2$ [32].

Many modifications to influence the mechanism, kinetics, and visible light activity of $TiO_2$ have been reported [33,34]. Other photocatalysts such as iron oxides [35] and zinc oxide (ZnO) [36] have also shown promising photocatalytic activity when compared with $TiO_2$. Other researchers have implemented various catalysts for photocatalytic OMW treatment. El Hajjouji et al. used $TiO_2$ under UV irradiation on the laboratory scale. The COD, coloration at 330 nm, and total phenols all decreased after a 24-h treatment to 22%, 57%, and 94%, respectively [20]. N-doped $TiO_2$ and undoped $TiO_2$ were prepared, characterized, and tested for degradation of organic content in OMW. The results show that doping with nitrogen produced higher catalytic activity [37]. Other photocatalysis systems have been used for OMW treatment; Nogueira et al. developed a combined approach consisting of two nanocatalysts ($TiO_2$ and $Fe_2O_3$) and biological degradation via fungi, and the percent removals of aromatic content, COD, and TPCs were 14%, 38%, and 31% respectively [38]. Sponza and Oztekin achieved the removal of dissolved COD, TPCs, and total aromatic amines of 99%, 89%, and 95% respectively by using $ZrO_2$-doped $TiO_2$ nanocomposite with a 14% mass ratio at different experimental conditions [39]. Graphene-$TiO_2$ was tested in the treatment of olive mill effluents via photocatalytic oxidation using different concentrations, contact time, and pH values; the pollutant removal efficiencies were 88%, 92%, and 95% for COD, TPCs, and TS respectively [40].

In our research group, many chemical, physical, and biological methods have been investigated to treat OMW; a review was published by Al Bawab et al. in 2017 summarizing many conducted studies in the Mediterranean region that investigated different OMW treatment methods [41]. Odeh et al. used a new type of surfactant, sodium polypropylene oxides sulphate of the type (branched hydrocarbon chain) (propoxyle group)-(sulphate) to enhance the OMW remediation using a modeled sample of OMW [42]. Al-Bawab et al. used modified extended surfactant (sodium polypropylene oxide sulfate combined with cationic hydrotropes tetra butyl ammonium bromide (TBAB) and the recovery of phenols using different combinations reached up to 99.8% [43]. In addition, cost-effective

media of two types of granular carbons were prepared to provide an economical treatment strategy in which the percentage removal of phenolic compounds from OMW samples was significantly enhanced by using oxidized granular activated carbon (GAC-OX) impregnated with Span 20 soaked for 15 days, yielding a percentage removal of 97.94 ± 2.30% [44].

In the field of nanotechnology, volcanic tuff-magnetite nanoparticles coupled with coagulation-flocculation methods were successfully implemented as a treatment approach for OMW real samples, in which the removal efficiency of volcanic tuff for COD was increased by acid activation followed by calcination. Further enhancement was achieved by coupling with 0.5% by weight of magnetite nanoparticles with maximum COD removal of 76% at pH 8. In addition, volcanic tuff activated by calcination and coupled with 0.5% by weight magnetite nanoparticles at pH 10 provided good removal for both TPC and COD of 73% and 70%, respectively [45]. Real OMW samples were also treated by using activated carbon (AC) prepared from olive cake waste and functionalized with $Cu/Cu_2O/CuO$ which offers a cost-effective treatment solution, the percent uptake of TPC was (85%), COD (42%), TSS (89%), and TDS (88%) by the adsorbent product [46]. Modified activated carbon (GAC) was tested in removing phenolic compounds from olive mill wastewater under acidic and basic conditions and in the presence and absence of different surfactants. The maximum removal of the phenolic compound was found to be 88% for reduced GAC at pH 9. However, the presence of surfactant(s) did not enhance the capacity of GAC [47].

Polymer membrane(s) impregnated with carbon nanotubes (CNTs) were developed, characterized, and evaluated for removing phenolic compounds from olive mill wastewater. The prepared membranes (PES/CNTs) with 0.5 wt. % CNTs showed the highest total phenol removal of 74% and the removal was enhanced by increasing the dose of CNTs [48].

Our research group also developed doped and undoped novel, nano-sized oxide photocatalysts with a fluorite type structure ($A_2B_2O_7$), characterized, and tested for removing dyes from textile wastewater. Results showed that the optimum dye degradation conditions were 1.5 g/L catalyst dose and pH 11. $Fe_2Zr_{0.85}W_{0.15}O_7$ showed promising photocatalytic activity for real textile wastewater, where the 69% COD removal was obtained under the same conditions used for methylene blue degradation [49]. Then Ga and Zr were the start points, where fluorite-type Zr-based oxides $Ga_2Zr_{2-x}W_xO_7$ (x: 0, 0.05, 0.015) were prepared, and the obtained results (pH 9, 1 g/L catalyst dose, and 300 min contact time) showed 83.6% photocatalytic decolourization of crystal violet [50]. Following the synthesized novel nano-sized $Ga_{2-x}Cu_xZr_{2-x}W_xO_7$ (x: 0, 0.05, 0.015) system, the photocatalytic activity of $Ga_{1.85}Cu_{0.15}Zr_{1.85}W_{0.15}O_7$ recorded 93.84% degradation of malachite green dye with 40 ppm malachite green dye in 300 min at pH 9 [51]. In another approach, fluorite-type $Fe_{2-x}Cu_xZr_{2-x}W_xO_7$ (x: 0, 0.05, 0.015) nanoparticles were prepared, and complete removal of 20 mg/L carbol fuchsin dye was achieved under optimal conditions (pH 9, and catalyst loading of 1.5 g/L) [52].

In this present study, three of these photocatalysts along with other novel photocatalysts with the fluorite-type structure were chosen to be prepared by a reliable, eco-friendly, and easy method to optimize the shape and grain size of nano-sized metal oxides. The prepared photocatalysts were characterized and tested in various pH conditions and at various time scales for photocatalytic degradation of phenolic compounds from olive mill wastewater.

## 2. Results

### 2.1. The Chemical Structure of the Prepared Nano Photocatalysts

Four nano photocatalyst systems were prepared: undoped $M_2Zr_2O_7$ (where M = iron (Fe), manganese (Mn), and copper (Cu)), doped $Fe_{2-x}Mn_xZr_2O_7$, doped $Fe_{2-x}Cu_xZr_2O_7$, and doped $Fe_2Zr_{2-x}W_xO_7$ (where x = 0.15 and 0.3). Table 1 shows the chemical structure of the prepared materials.

**Table 1.** The chemical structure and the abbreviation of the prepared nano photocatalysts.

| Photocatalyst System | Photocatalyst Structure | Abbreviation |
|---|---|---|
| $M_2Zr_2O_7$ | $Fe_2Zr_2O_7$<br>$Cu_2Zr_2O_7$<br>$Mn_2Zr_2O_7$ | FeZr<br>CuZr<br>MnZr |
| $Fe_{2-x}Mn_xZr_2O_7$ | $Fe_{1.85}Mn_{0.15}Zr_2O_7$<br>$Fe_{1.70}Mn_{0.30}Zr_2O_7$ | FeMn1<br>FeMn2 |
| $Fe_{2-x}Cu_xZr_2O_7$ | $Fe_{1.85}Cu_{0.15}Zr_2O_7$<br>$Fe_{1.70}Cu_{0.30}Zr_2O_7$ | FeCu1<br>FeCu2 |
| $Fe_2Zr_{2-x}W_xO_7$ | $Fe_2Zr_{1.85}W_{0.15}O_7$<br>$Fe_2Zr_{1.70}W_{0.30}O_7$ | ZrW1<br>ZrW2 |

*2.2. Characterization of the Prepared Nano Photocatalysts*

The XRD patterns of FeZr, MnZr, and CuZr photocatalysts consist of a defect fluorite-type structure ($A_2B_2O_7$) due to the ratio of ionic radius of cations; when the ratio $r_A/r_B$ is less than 1.46 or more than 1.78, the defect-fluorite structure will be formed [53]. In the present study, $r_A/r_B$ ratios were found to be 0.76, 1.11, and 1.101 for FeZr, MnZr, and CuZr respectively (r values for $Fe^{3+}$, $Mn^{2+}$, $Cu^{2+}$, and $Zr^{4+}$ are 55, 80, 73, and 72 pm, respectively [54]). Since most of the prepared materials are novel and do not have ICDD cards, the best match was found to be with $La_2Zr_2O_7$ (ICDD code 00-017-0450) for all prepared photocatalysts (except for the CuZr XRD pattern), which confirms the defect fluorite-type crystal structure for the prepared materials. The shift to a higher 2Θ value for the prepared materials is related to the difference in ionic radii with an La ion. The 2Θ peaks of prepared photocatalysts at 31°, 36°, 51°, and 60° correspond to crystal planes of [111], [200], [220], and [311], respectively (Figure 1) [49,51,55,56]. CuZr is a combination of defect fluorite structure as a major phase (as expected from the value of $r_A/r_B$ ratio) and monoclinic crystal structure as a minor phase since it matches with $La_2Zr_2O_7$ and CuO (ICDD code 00-001-1117) [57].

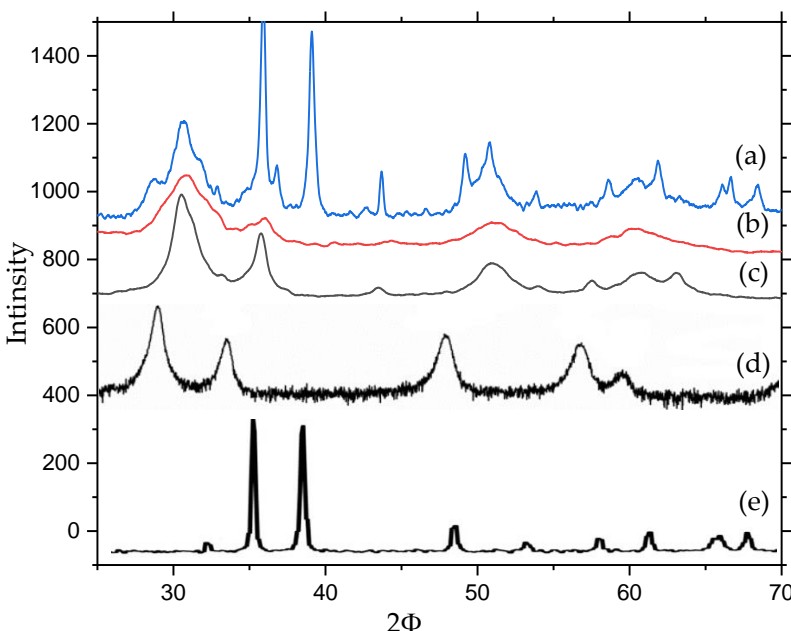

**Figure 1.** Powder x-ray pattern of (a) CuZr, (b) MnZr, (c) FeZr, (d) $La_2Zr_2O_7$ (ICDD code 00-017-0450), and (e) CuO (ICDD code 00-001-1117).

The broad peaks of the XRD (Figures 1 and 2) results indicate that all the prepared materials are nanosized, and the peak shifts to a higher 2Θ shown in Figure 2 confirming

the success of the doping of Fe by Mn, Cu, and W, respectively [56]. The increase in 2Θ shift after doping can be explained according to Bragg's equation, which indicates that increasing dopant concentration which also exhibits a larger ionic size decreases the crystal spacing (d) then the position of 2Θ will increase [57].

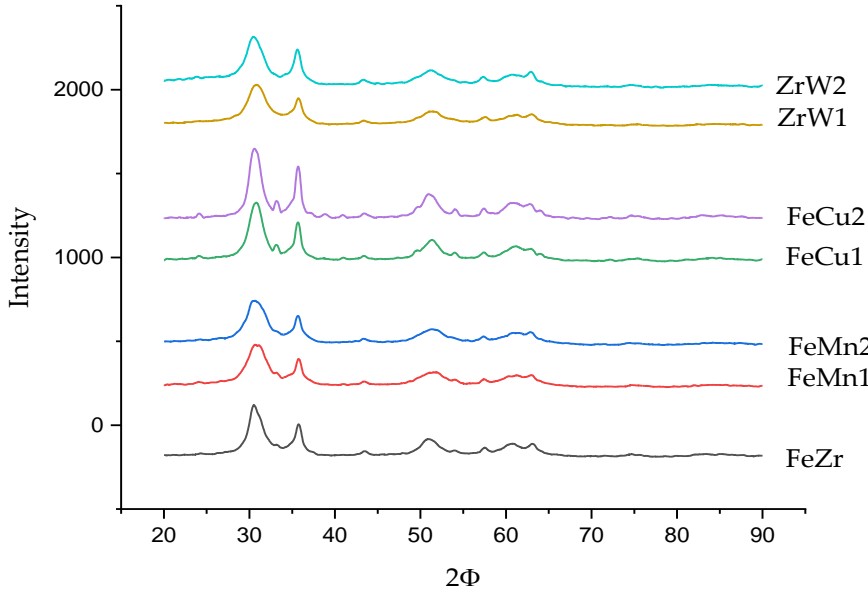

**Figure 2.** Powder x-ray pattern for the prepared doped systems.

The average particle size was measured by Scherrer's equation:

$$D_p = K\lambda/(B \cos\Theta), \tag{1}$$

where: $D_p$: crystallite size; $K$: Scherrer constant. Note that $K$ varies from 0.68 to 2.08, and $K = 0.94$ for spherical crystallites with cubic symmetry. λ: X-ray wavelength. For XRD, the Cu Kα average = 1.54178 Å. $B$: Full width at half maximum (FWHM) of the XRD peak. Θ: XRD peak position where $\Theta = 2\Theta/2$.

The measured average particle size of the prepared photocatalysts shown in Table 2 indicates that doping Fe with Mn or Cu increases particle size, but doping Zr by W decreases particle size. Additionally, increasing the dopant concentration increases the particle size due to the higher dopant size [49,58].

**Table 2.** Average particle size of the prepared nano photocatalysts.

| Photocatalyst | Average Particle Size (nm) |
|:---:|:---:|
| FeZr | 6.40 |
| CuZr | 17.0 |
| MnZr | 7.80 |
| FeMn1 | 6.10 |
| FeMn2 | 6.80 |
| FeCu1 | 6.90 |
| FeCu2 | 7.40 |
| ZrW1 | 5.90 |
| ZrW2 | 6.00 |

To approve the composition of the prepared photocatalysts, ICP-OES analysis was used to determination of the mole ratio (A/B; A: Fe, Mn, or Cu elements and B: Zr). The results are in agreement with the expected mole ratio of FeZr, MnZr, and CuZr as shown in Table 3.

**Table 3.** ICP-OES analysis for the $M_2Zr_2O_7$.

| Parent Photocatalyst | Theoretical Formula | Theoretical Mole Ratio (A/B) | ICP-OES Results (ppm) | mol/L | Experimental Mole Ratio |
|---|---|---|---|---|---|
| FeZr | $Fe_2Zr_2O_7$ | 1 | Fe = 1963.69<br>Zr = 3189.10 | 0.03516<br>0.03496 | 0.99 |
| CuZr | $Cu_2Zr_2O_7$ | 1 | Cu = 6521.50<br>Zr = 9005.32 | 0.10260<br>0.09872 | 1.04 |
| MnZr | $Mn_2Zr_2O_7$ | 1 | Mn = 1150.90<br>Zr = 1911.03 | 0.02095<br>0.02083 | 1.01 |

TEM imaging of the prepared photocatalysts is shown in Figures 3 and 4. The morphology consists of small quasi-spherical particles in the nano-scale range. Nanoparticle aggregates up to 35 nm are also observed. The aggregation of nanoparticles in MnZr is the lowest which makes the effective surface area for degradation of phenols the highest. This agrees with the literature [59]. The average particle size measured by TEM is larger than that of XRD this is due to the aggregation [60]. It is notable that the aggregation is the least in MnZr, which makes it the most effective photocatalyst that has the highest effective surface area.

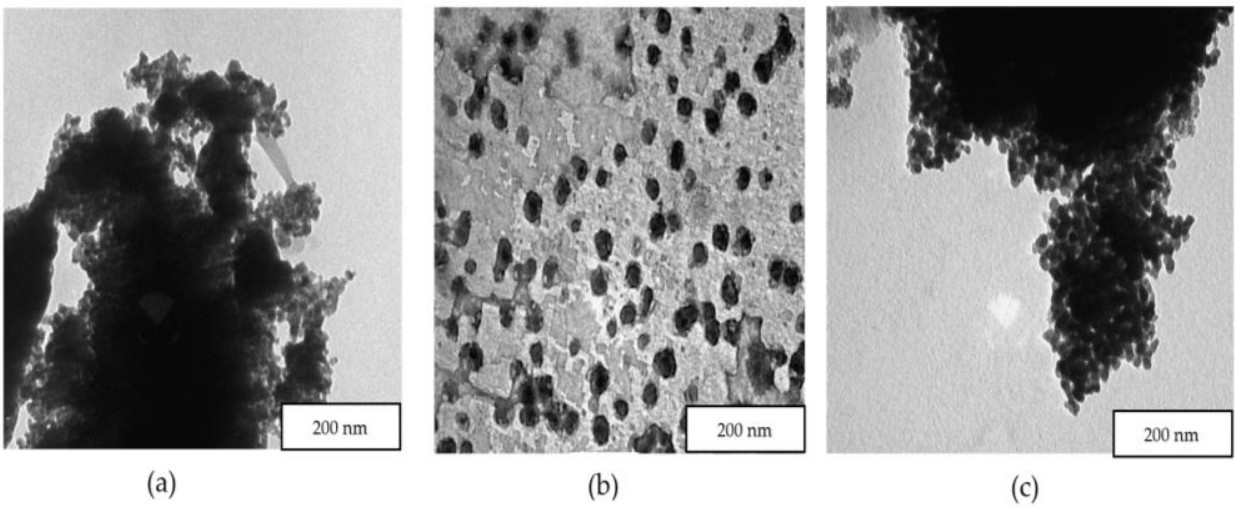

**Figure 3.** TEM images of (**a**) FeZr, (**b**) MnZr, and (**c**) CuZr.

To determine the surface area of the prepared photocatalysts, *the Brunauer-Emmett-Teller (BET) method* using nitrogen physisorption was used. The pore volumes were calculated with the desorption data from adsorption-desorption isotherms (based on *Barrett–Joyner–Halenda (BJH) theory).*

The surface area of FeZr is 67 m$^2$/g. Doping Fe with Mn increased the surface area to 173 m$^2$/g. Increasing the dopant concentration slightly increased the surface area to 187 m$^2$/g this is provided by the results of photocatalytic degradation which is the highest in FeMn2.

For MnZr, CuZr, and FeZr the degradation activity is highest in MnZr followed by CuZr then FeZr. Of these three photocatalysts, MnZr had the lowest BET surface area (Table 4). To explain why MnZr exhibits the highest photocatalytic degradation but has the lowest BET surface area; the point of zero charges ($P_{ZC}$) was measured via the pH drift method (Figure 5). The simplest compound of phenolic compounds is phenol, which has a pH value of 10. The pH of other phenolic compounds is changeable depending on the groups attached to the benzene ring; electron-donating groups increase the pH by more than 10 and electron-withdrawing groups decrease the pH by less than 10. Any phenolic

compound, is protonated under its pH and is in the form of a phenoxide ion above its pH. Due to the values of $pH_{pzc}$, they were found to be equal to 6, 5.8, and 8.2 for MnZr, CuZr, and FeZr, respectively; the number of phenolic compounds in the protonated form is higher in the negative surface of MnZr and CuZr than in the case of FeZr.

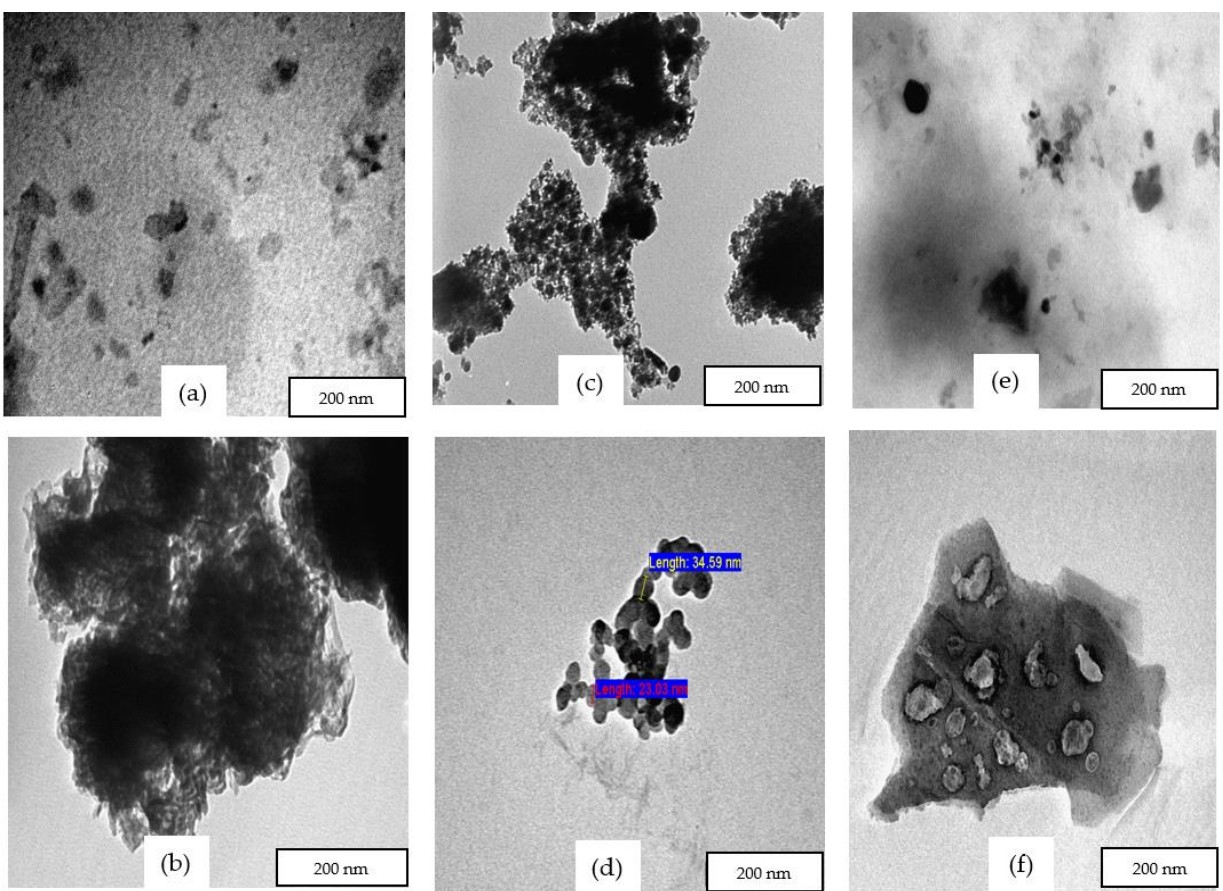

**Figure 4.** TEM images of doped systems (**a**) FeMn1, (**b**) FeMn2, (**c**) FeCu1, (**d**) FeCu2, (**e**) ZrW1, and (**f**) ZrW2.

**Table 4.** The values of the surface area, total pore volume, and average pore diameter of prepared photocatalysts.

| Photocatalyst Structure | Surface Area (m$^2$/g) | Total Pore Volume (cc/g) | Average Pore Diameter (A$^\circ$) |
|---|---|---|---|
| MnZr | 30 | 0.0220 | 29.182 |
| CuZr | 62 | 0.0661 | 42.992 |
| FeZr | 67 | 0.1204 | 71.672 |
| FeMn1 | 173 | 0.1615 | 37.376 |
| FeMn2 | 187 | 0.1585 | 33.988 |

$P_{zc}$ values of MnZr and CuZr are close, which means that more than 5.8 the surface is negatively charged and for FeZr the surface is negatively charged more than 8.2. It is believed that the higher negative charged surface can attract the protonated form of phenolic compounds through strong electrostatic attraction forces then increases the photocatalytic degradation activity [61].

The leaching experiment was done by mixing the prepared photocatalyst with deionized water at pH, contact time, dose, and condition identical to the reaction media, and it found that the concentrations (in ppm) are very small or not detectable; for FeZr: [Fe$^{2+}$] is

0.046 and $[Zr^{4+}]$ is 0.401, for MnZr: $[Mn^{2+}]$ is 0.445 and $[Zr^{4+}]$ is N.D., and for CuZr: $[Cu^{2+}]$ is 0.0592 and $[Zr^{4+}]$ is N.D.

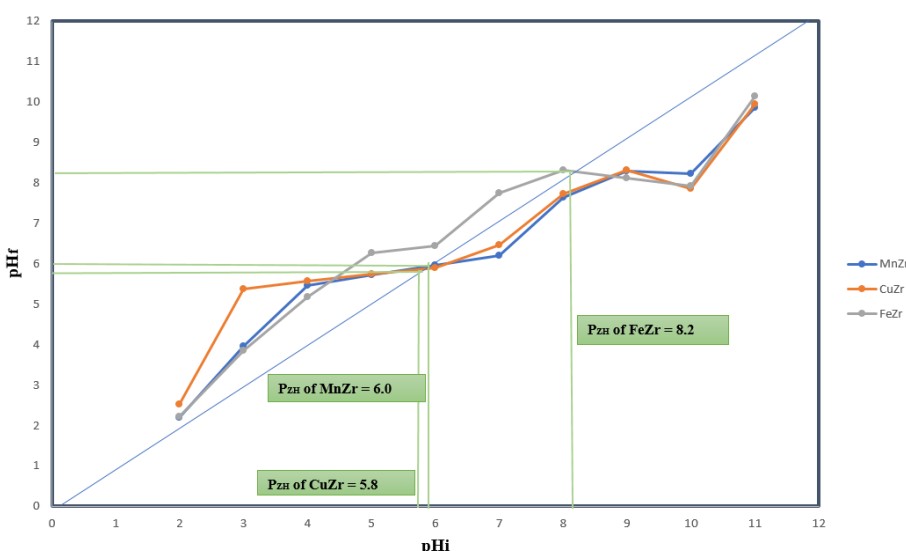

**Figure 5.** The measured $P_{ZH}$ of MnZr, CuZr, and FeZr.

### 2.3. Photocatalytic Activity of the Prepared Material on OMW Treatment

The prepared photocatalysts were used for OMW treatment by testing the photocatalytic degradation of total phenolic compounds. The percent degradation of total phenolic compounds was measured using the equation:

$$\%Degradation = \frac{C_i - C_e}{C_i} \times 100 \tag{2}$$

where $C_i$: total phenolic compound concentration before degradation; $C_e$: total phenolic compound concentration after degradation.

The effects of contact time and pH were examined as follows:

Increasing contact of the prepared photocatalysts and OMW under light was studied and all were found to have the same behavior. It is clear that the degradation activity rapidly increases in the first 6 h of exposure; then the degradation is slowly increased up to up to 24 h, after which it is plateaued due to an increase in the excitation sites at first, then with time, the catalysts were consumed. Figure 6 shows the effect of contact time in the $M_2Zr_2O_7$ systems. Figure 7 shows these effects in the doped systems.

Many factors control the photocatalytic degradation of total phenols, such as time of exposure, pH, photocatalyst composition, and light strength. Nogueira et al. studied the photocatalytic oxidation of various photocatalyst combinations using a concentration of 1.0 g/L. As part of the study, the percent removal of TPCs during overnight UV light exposure was measured. The results showed that introducing $H_2O_2$ to the photocatalysts significantly enhanced TPC removal. The measured percent removal for $TiO_2$, $Fe_2O_3$, $TiO_2/H_2O_2$, and $Fe_2O_3/H_2O_2$ was 5.5, 9.9, 31.2, and 25.5, respectively [62]. Baransi et al. combined $TiO_2$-$Fe_2O_3$ under 60 W/UV radiation to remove up to 38% of TPCs using 15 mg of the material in 1 L of OMW [63]. Ugurlu and Karaoglu prepared $TiO_2$ supported on nano sepiolite and reported up to 61% removal of TPCs by increasing the exposure time to 50 min [64]. Increasing the incident UV light has a large impact on TPCs removal. By using 300 W UV light power, Sponza and Oztekin removed up to 89% of TPCs within 60 min using 15 mg of $ZrO_2/TiO_2$ nanocomposite in 1 L of OMW [39]. Graphene-$TiO_2$ was tested in treating of OMW and efficiencies of pollutants removal were 88%, 92%, and 95% for COD, TPC, and TS, respectively, within 30 min under 300 W/UV [40].

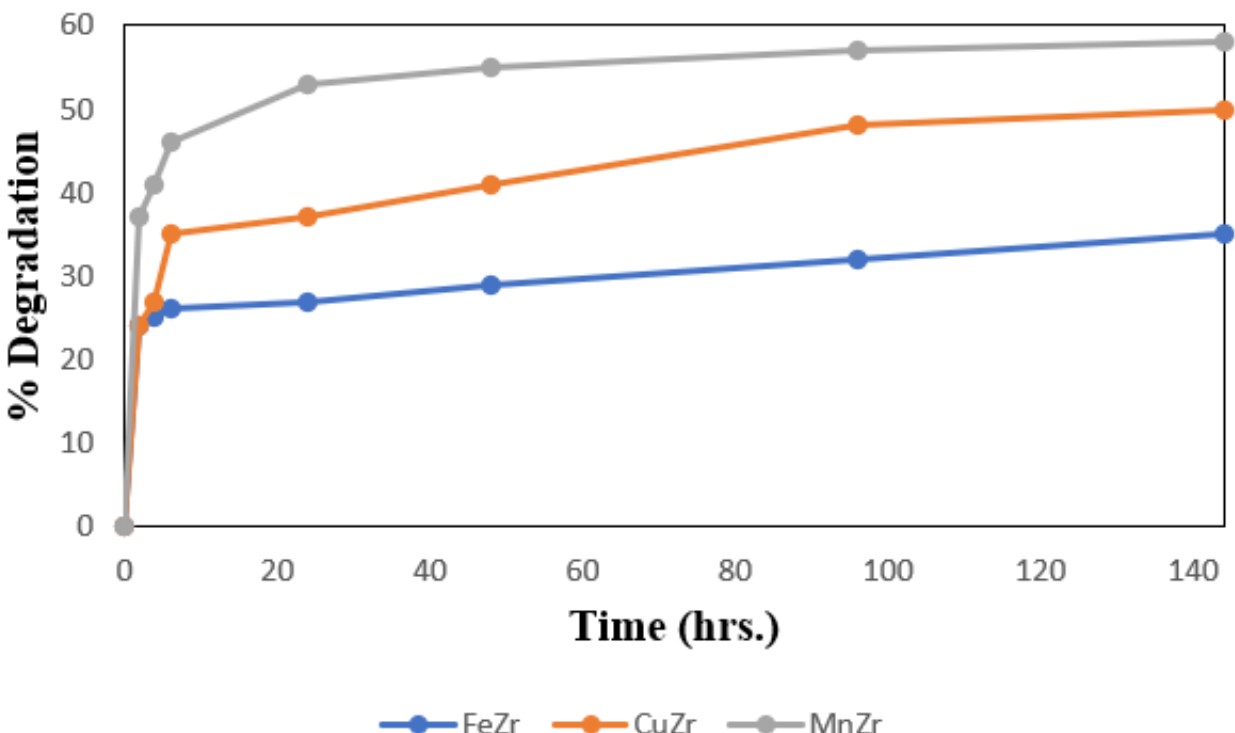

**Figure 6.** Contact time effect ranging from 2 h to 6 days at pH 6 for $M_2Zr_2O_7$ systems in OMW under the light.

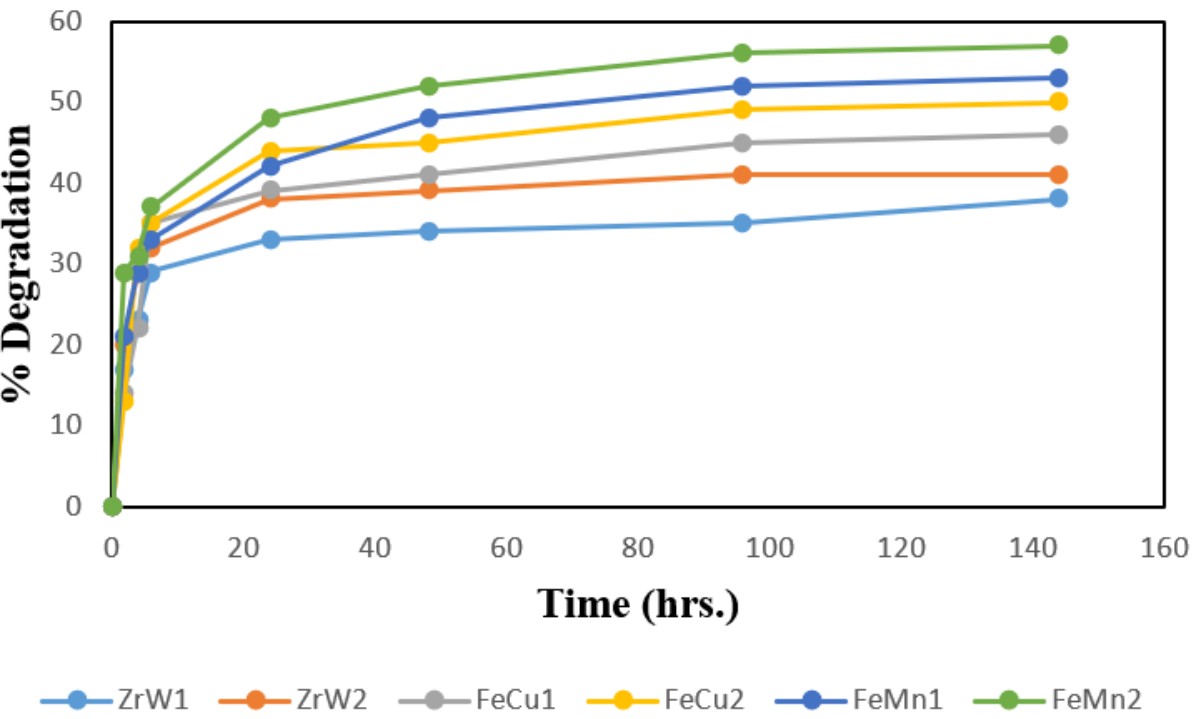

**Figure 7.** Contact time effect ranging from 2 h to 6 days at pH 6 for $M_2Zr_2O_7$ doped systems in OMW under the light.

As shown in Figure 8, increasing pH increases the photocatalytic degradation of phenolic compounds. The literature provides several explanations of how pH can affect the degradation efficiency of heterogeneous catalysts. Badawy et al. found that increasing the OMW pH from 2 to 10 increased TPC removal from 38.4% to 70.1% within one hour

of exposure time. They attribute this observation to the fact that the intermediate of any AOPs contains OH radicals (Figure 9), which are produced from the reaction between $OH^-$ and a positive hole. Basic medium enhances this reaction, thus enhancing the degradation activity [65]. Chan and Chu suggested that nanomaterials preferentially aggregate at low pHs, decreasing photocatalytic activity [66]. $TiO_2$ supported on nano sepiolite also achieves optimum removal at pH 9–10 [64].

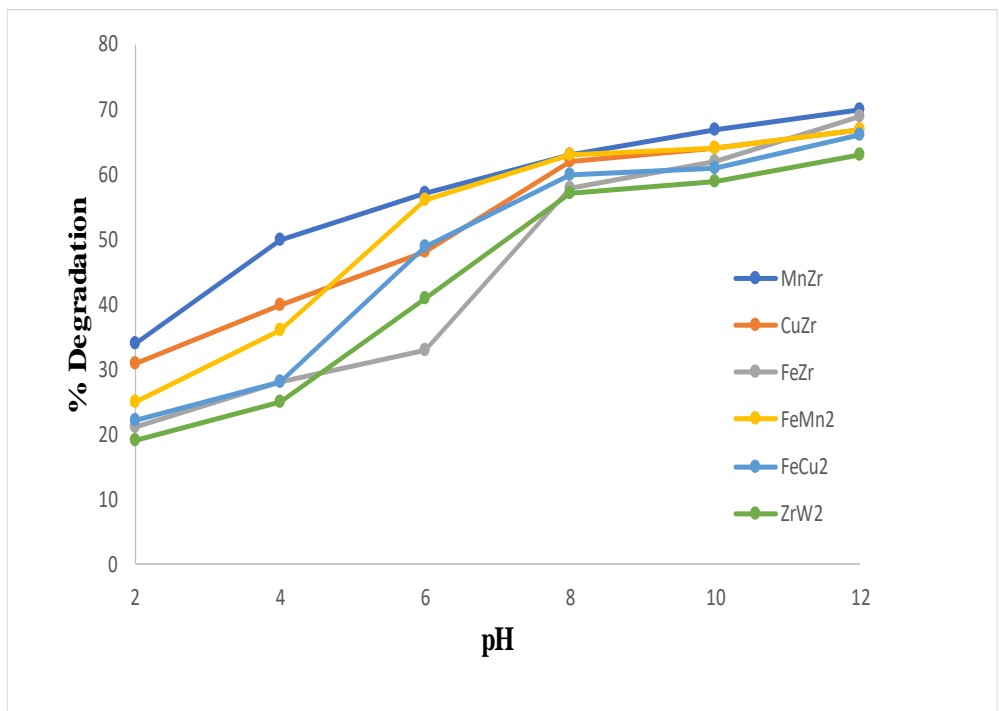

**Figure 8.** pH impact on photocatalytic degradation of phenolic compounds using doped and undoped systems for 1 day contact time; pHs ranging from 2 to 12.

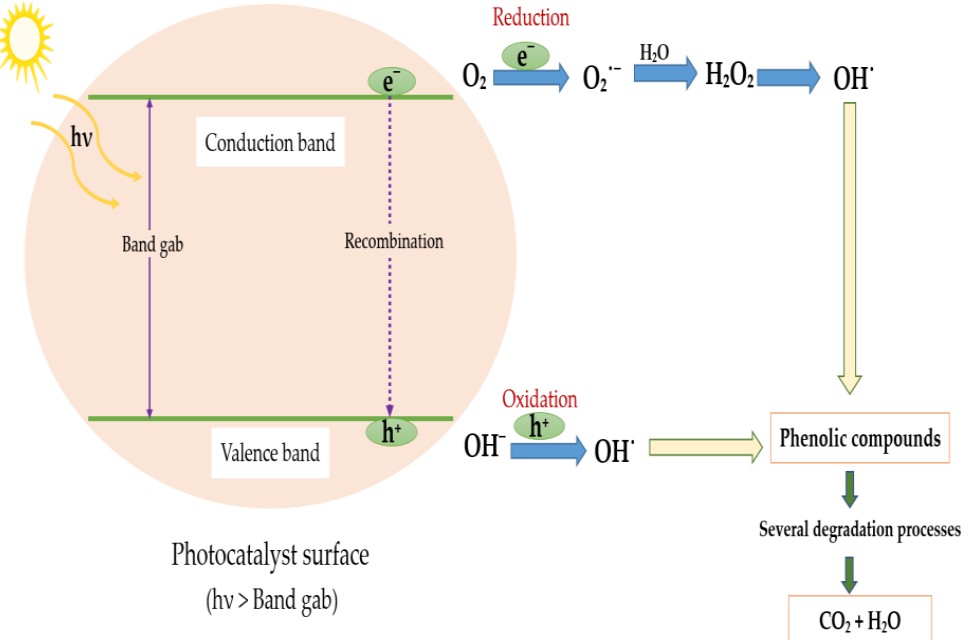

**Figure 9.** Schematic of the photocatalytic degradation mechanism.

## 3. Materials and Methods

### 3.1. Materials

Iron (III) nitrate nonahydrate (Guangdong Guanghua Sci-Teach Co., Ltd., China), zirconium (IV) oxynitrate hydrate (Sigma Aldrich, St. Louis, MO, USA), manganese (II) chloride tetrahydrate (Peking Chemical Works, Beijng, China), copper (II) nitrate trihydrate (Scharlau Chemie S. A., Sentmenat, Barcelona, Spain), tungsten (VI) chloride (Sigma Aldrich, Chemie GmbH, Taufkirchen, Germany), citric acid anhydrous extra pure (Loba Chemie, Colaba, Mumbai, India), ethylene glycol (ROAD, Sandy Croft, Deeside, CLWYD, China), hexane (Biosolve Chimie SARL, Dieuze, France), ethyl acetate (Carbon Group, Elmstead Market, Essex, CO7 7FD, England), methanol (Biosolve Chimie, SARL, Dieuze, France), Folin-Ciocalteu reagent (Biosolve Chimie SARL, Dieuze, France), sodium carbonate solution (analytical reagent grade, FISCHER Chemicals, Guangzhou, China), and gallic acid (Alfa Aesar, Kandel, Germany) solvents and reagents were analytical grade and used without further purification.

### 3.2. Preparation of Photocatalysts

Nano-sized systems were prepared via the *pechini method* [67] using citric acid (CA) as a chelating agent with dissolved polymeric precursors (Figure 10). These chelates formed water and ester by reacting with ethylene glycol (EG). Heating was used to obtain a gel, which then forms a powder of desired stoichiometry via thermal decomposition.

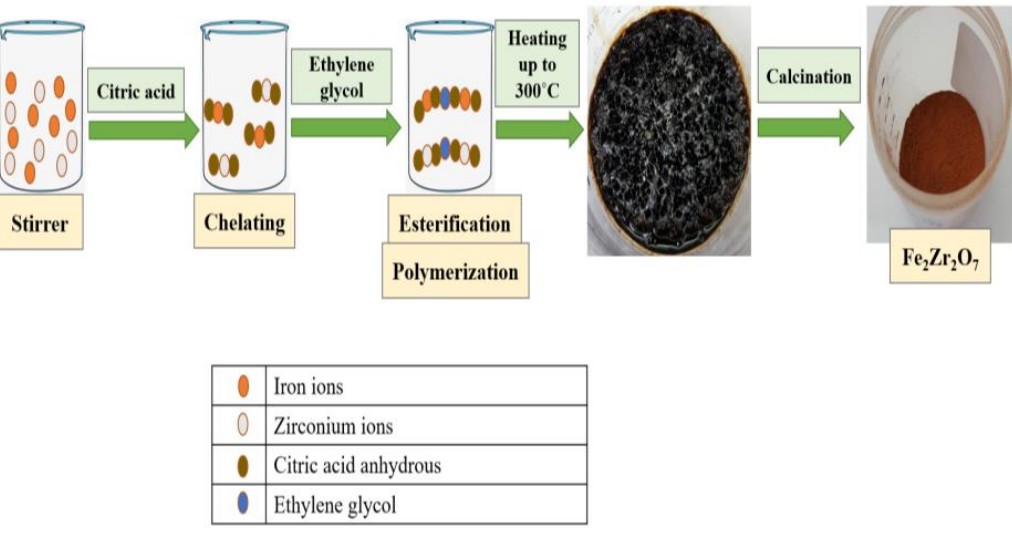

| | |
|---|---|
| ● | Iron ions |
| ○ | Zirconium ions |
| ● | Citric acid anhydrous |
| ● | Ethylene glycol |

**Figure 10.** Pechini procedure for preparation of $Fe_2Zr_2O_7$ photocatalyst.

Doping was performed for $Fe_2Zr_2O_7$ photocatalysts using manganese (Mn) and copper (Cu) as dopants for iron (Fe) and tungsten (W) as a dopant for zirconium (Zr) by substituting x moles of the dopant before CA addition.

The systems were prepared as follows: the precursor metal salts, CA (M molar ratio of 4:1), and EG (M molar ratio of 1:1.5) were mixed according to the suitable stoichiometry to achieve the desired oxide solution. The temperature was gradually increased under continuous stirring to induce esterification and polymerization until gel was obtained. The gel was gradually heated to 300 °C, then ground and calcined in a muffle furnace at 200 °C for 2 h, followed by 300 °C for 2 h, then 500 °C for 2 h to form the final powder.

Figure 11 shows various preparation stages of the $M_2Zr_2O_7$ system, which starts as a solution, then forms a gel, followed by a powder after calcination.

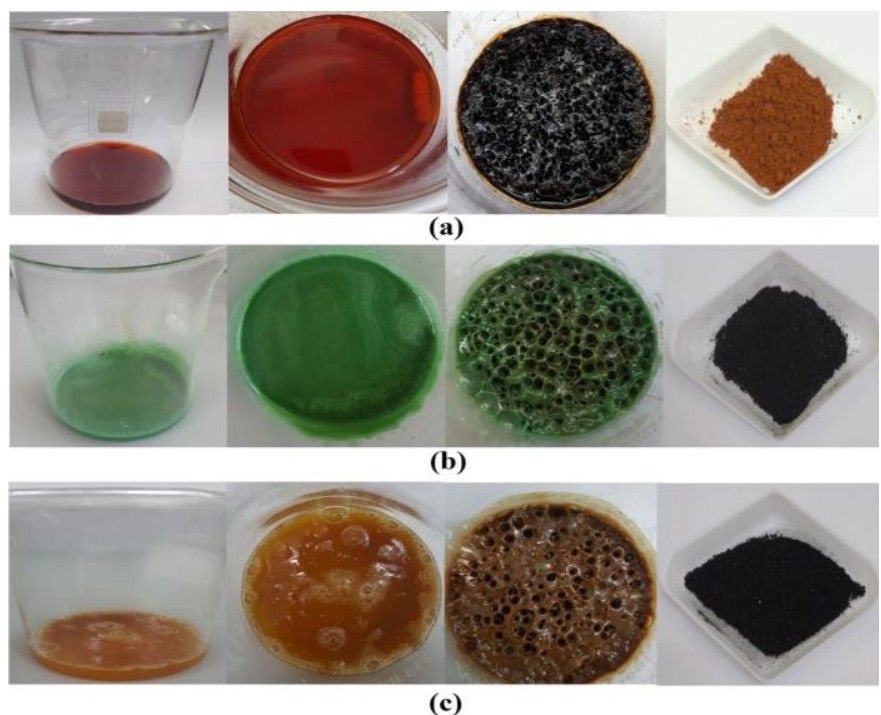

**Figure 11.** Stages for preparation of $M_2Zr_2O_7$ system; starting with the solution, gel, ceramic, then powder nanomaterial, (**a**) FeZr, (**b**) CuZr, and (**c**) MnZr.

### 3.3. Characterization of the Prepared Material

To identify the crystal structure and measure the particle size and lattice parameters of the prepared materials, XRD was used. XRD was carried out using a 7000 Shimadzu 2 kW model spectrophotometer with a nickel filtrated Cu radiation (CuK$\alpha$) with $\lambda$ = 1.54056 Å. The 2$\Theta$ range of scans was 20–90 with a 0.02 step size. In order to confirm the composition of the prepared materials and measure the leached metals from the reaction media, if any, inductively coupled plasma/optical emission spectroscopy (ICP-OES) (GBC E1475, Hampshire, MA, USA) was used; the samples were conducted to acid digestion first by mixing 0.05 g of the sample with a mixture of $NHO_3$ (5 mL) $H_2O_2$ (5 mL), and HF (2 mL) concentrated acids then heating up to 100 °C for 3 h. The microstructure (i.e., morphology, shape, and size) of the prepared photocatalysts were studied using TEM imaging (MORGAGNI 268-FEI company, Eindhoven, Netherlands). The specific surface area, average pore diameter, and total pore volume were measured using a BET Surface Area Analyzer (Quantachrome NOVA 2200E BET, Boynton Beach, FL, USA).

### 3.4. OMW Pretreatment

The OMW was initially submitted to a coagulation/flocculation process to reduce the amount of total suspended solids (TSS). OMW samples were pretreated in a jar test apparatus (VELP SCIENTIFICA JLT6, Usmate (MB), Italy) equipped with stainless steel stirring impellers and six 1-L beakers. This was used according to a previously reported procedure [45] as follows: the pH of 500 mL of OMW was adjusted to 7 under rapid mixing (200 rpm) for 6 min then a 7% solution of ferric chloride (Hangzhou Soya Co. Ltd., Zhejiang, China) was added as a coagulant. Next, polymeric quaternary amine (SUPERFLUC C 577, KERMIRA, Helsinki, Finland) was added as a flocculant after 20 min at 30 rpm. Finally, mixing was stopped to allow for aggregation settling.

### 3.5. Evaluation of Photocatalytic Activity

The prepared nano-sized materials have been tested for OMW treatment using a batch approach. A set quantity of prepared materials was mixed with pretreated OMW (0.05% material: OMW), and the system was maintained under stirring using a commercial LED

lamp (45 W) for variable contact time (2 h–6 days) and pH (2–12). The supernatant solution was separated, and total phenolic compounds (TPCs) were measured.

### 3.6. Spectrometric Measurement of Total Phenolic Compounds (TPCs)

TPCs were measured before and after OMW treatment via the Folin-Ciocalteu method using gallic acid as calibration standard [68] as follows: the pH of a 2 mL sample of treated OMW was adjusted to 2 then defatted with hexane (1:1 $v/v$) for 15 min and centrifuged for 10 min at 4500 rpm. The TPCs were double extracted from the defatted sample using ethyl acetate (1:1 $v/v$) for 15 min, separated, and evaporated at 60 °C using a rotary evaporator (RE 300, MESLO, Cyprus). Ten milliliters of methanol was added to the residue and a 1-mL aliquot (or gallic acid standard) was added to 9 mL of deionized water. The solution was colored using a Folin-Ciocalteu reagent (1:1 $v/v$), and 10 mL of 7% sodium carbonate solution was added. The total solution volume was brought up to 25 mL using deionized water. The final solution was kept in the dark for 120 min; then the absorbance at $\lambda_{max}$ = 750 nm was measured using a Spectro Direct- Lovibond single-beam (UK instrument). Figure 12 shows $\lambda_{max}$ = 750 nm for various concentrations of gallic acid standards.

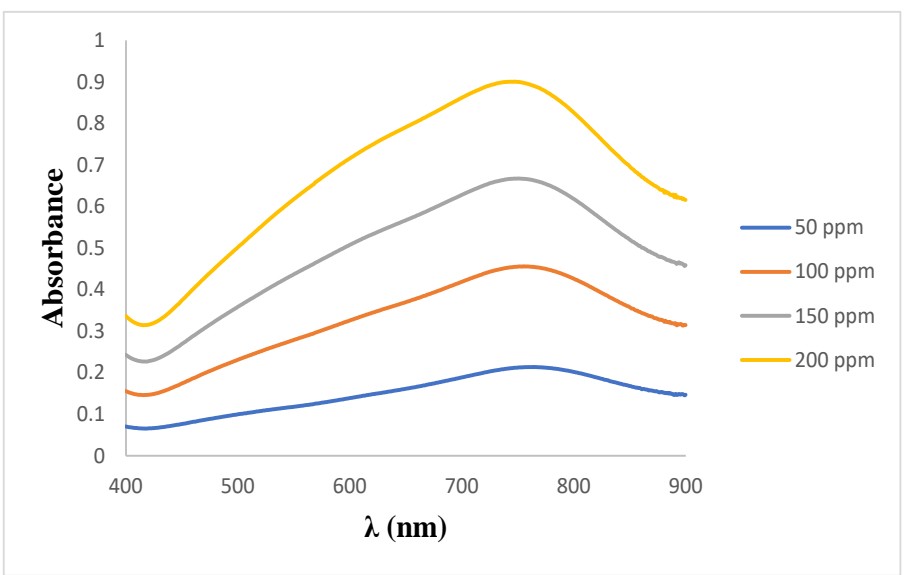

**Figure 12.** Absorbance as a function of wavelength showing $\lambda_{max}$ at 750 nm.

### 4. Conclusions

Novel nano-photocatalysts were prepared and their photocatalytic degradation of total phenolic compounds was tested. XRD and TEM analysis confirm the nanoscale of all prepared and doped materials (5.9–17.8 nm). XRD and ICP-OES analysis confirms the defect fluorite type crystal and the composition of the prepared photocatalysts. The most effective photocatalyst for phenol degradation was $Mn_2Zr_2O_7$. Studying pH impacts on the degradation process confirms that the photocatalyst system performed most effectively in basic medium. The TPCs removal reached up to 70% using a basic medium up to pH 10 and a long exposure time up to one day. Preparing adsorbent-$Mn_2Zr_2O_7$ composites and increasing the LED lamp intensity can provide a promising OMW treatment approach.

**Author Contributions:** Conceptualization, A.A.B., M.A.-D. and A.K.; methodology, D.A.-D. and A.K.; validation, A.A.B. and M.A.-D.; formal analysis, A.A.B., M.A.-D. and D.A.-D.; investigation, D.A.-D.; resources, D.A.-D.; data curation, D.A.-D., A.A.B. and M.A.-D.; writing—original draft preparation, D.A.-D.; writing—review and editing, A.A.B., M.A.-D., A.K. and D.A.-D.; visualization, A.K.; supervision, A.A.B. and M.A.-D.; project administration, A.A.B. and M.A.-D.; funding acquisition, A.A.B. and M.A.-D. All authors have read and agreed to the published version of the manuscript.

**Funding:** This research was funded by Middle East Desalination Research Center (MEDRC) (financial project number 18-PJ-01).

**Acknowledgments:** Authors would like to acknowledge Middle East Desalination Research Center (MEDRC) for financing the project and the Deanship of Academic Research (DAR)/The University of Jordan, Amman, Jordan.

**Conflicts of Interest:** The authors declare no conflict of interest.

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
