# Peer review of "Olive Mill Wastewater (OMW) Treatment Using Photocatalyst Media"

_catalysts, doi:10.3390/catal12050539_

Round 1

Reviewer 1 Report

The novelty of the work could not be verified since lack of adequate characterization. How did the authors confirms the structure of the synthesized nanophotocatalyst? Either they could use XPS, ICP or any other quantitative technique to conclude the structure. XRD technique used to characterize the nanophotocatalyst is not helping to reach the structural conclution. XRD pattern in Figure 1 corresponds to monoclinic structure of CuO. How a cubic fluorite structure could be matched with a monoclinic crystal structure. Instead, the authors should select the pattern from ICDD pattern code: 00-016-0799. As per the information drawn from the XRD pattern, there is no evidence of A2B2O7 formation. Mostly looks like some mixed metal oxides.

The second major technique used to estimate the morphology (TEM) was not clear. The scale bar itself could not visible.

I think the manuscript shoul improve a lot mainly with characterization. So I will suggest to reject  the current format and suggest the authors to add more supporting data to reach the structure confirmation of the as prepared nanophotocatalyst.  

Reviewer 2 Report

This work deals with the preparation of nanophotocatalysts of M2Zr2O7 (M=Mn, Cu, and Fe), doped with Fe2Zr2O7, via sol-gel, and used in the photocatalytic degradation of olive mill wastewater (OMW). The materials were characterized by XRD, TEM and BET.

I recommend its publication in Catalysts after a few points are addressed:

  • Abstract and more places: surface area values usually do not have decimal positions due to the low accuracy of the BET method to obtain these values. The authors can check this fact in the related literature where surface area values are reported.
  • Why a “(“ at the end of line 225?
  • Table 3 is repeated (in lines 211 and 234). The first apparition of Table 3 in line 211 is a mistake.
  • Figure 8 is also repeated (lines 225 and 236). Remove the first one since the second one has its caption figure.
  • Table 3: surface area values have 3 decimal positions!! Usually they do not have any decimal position as it has been commented before.
  • Figure 11: separate “%” and “Degradation” in the title of the Y axys.
  • Figure 11: it is better to show the experimental data as points in the figure linked by trend lines and not this way with straight lines and sudden changes in the slope of the lines.
  • Only phenolic compounds were analysed in the final effluents of the reactions. The OMW was submitted to an initial pretreatment by coagulation/flocculation to reduce the amount of total suspended solids. However, which other compounds can be present in the solution submitted to photocatalysis in addition of phenolic compounds? This OMW after this pretreatment was analyzed to check if only or the main part of the solution were these phenolic compounds? Can they interfere in the removal of these phenolic compounds? Competitive effects can be produced during the photocatalytic process among phenolic compound and other possible compounds in the reaction media?
  • Figure 15 is not mentioned in the text (it must be in line 354).
  • What about the leaching of the metals belonging to the catalysts to the reaction media? The concentrations of these metals were quantified in the final effluents?

Author Response

Reviewer 2

Thank you for all your valuable comments and questions that for sure enrich our manuscript and made it clearer and more understandable for the readers

No

Comments and Suggestions for Authors

Answer

Location of added sentence(s) in the manuscript text (( green highlight color)  Page / line

This work deals with the preparation of nanophotocatalysts of M2Zr2O7 (M=Mn, Cu, and Fe), doped with Fe2Zr2O7, via sol-gel, and used in the photocatalytic degradation of olive mill wastewater (OMW). The materials were characterized by XRD, TEM, and BET.

I recommend its publication in Catalysts after a few points are addressed:

Abstract and more places: surface area values usually do not have decimal positions due to the low accuracy of the BET method to obtain these values. The authors can check this fact in the related literature where surface area values are reported.

Surface area values were  corrected

1/20

11/250 and 251

12/275

Why a “(“at the end of line 225?

Letter “A” was deleted

11/263

Table 3 is repeated (in lines 211 and 234). The first apparition of Table 3 in line 211 is a mistake.

The first apparition of Table 3 in line 211 was deleted

11/247

Figure 8 is also repeated (lines 225 and 236). Remove the first one since the second one has its caption figure.

The first figure was removed

11/266

Table 3: surface area values have 3 decimal positions!! Usually they do not have any decimal position as it has been commented before.

Table 3: surface area values are corrected now

12/274

Figure 11: separate “%” and “Degradation” in the title of the Y-axis.

The “%” and “Degradation” in the title of the Y-axis were separated.

15/338

Figure 11: it is better to show the experimental data as points in the figure linked by trend lines and not this way with straight lines and sudden changes in the slope of the lines.

The experimental data are shown as points in the figure now

15/338

What about the leaching of the metals belonging to the catalysts to the reaction media? The concentrations of these metals were quantified in the final effluents?

Results of leaching of metals belonging to the prepared photocatalyst are introduced now

12/280

18/387

Thank you for all your valuable comments and questions that for sure enrich our manuscript and made it clearer and more understandable for the readers

Reviewer 3 Report

     The Authors provide an introduction to the subject matter, present the aim of the work. However, information in the introduction could be better organized to avoid turmoil, and the aim of the work should be better, more precisely defined.

     The presented work included the preparation and the use of novel materials for photocatalytic degradation of phenolic compounds from olive mill wastewater. The catalysts were characterized and tested in various pH conditions, and in various time scales of the degradation processes.

     The manuscript is quite well prepared. The language of the paper is rather decent. The results are discussed. However, in order to consider the submission as suitable to forward to the next step of the proceeding, I suggest implementation of the following amendments:

  • The language of the manuscript could be improved, especially in terms of syntax to provide better clarity of the text. The spacing also requires correction at several points of the text.
  • In the line 218: should be ‘degradation’ not ‘digradation’.
  • Figures: 9, 10 and 11 – a lack of basic data and parameters of the experiments in the captions of the Figures. The captions must provide detailed and complete set of information independently of description of the experiments in the text.
    Furthermore, in the Figures 9, 10 and 11 exact measurement points must also be included. The scatter chart is appropriate here not a line chart. To introduce and visualization the experimental data which are not measured continuously a line chart is unsuitable. In this form, the Figures 9, 10 and 11 are illegible.
  • The results of the experiments, presented in the Figures 9 and 10, must be discussed in detail in the text. These results, in fact, were not discussed at all. The absence of this discussion is a huge disadvantage of the submission, and seriously impoverishes a factual value of the manuscript because this part of the experimental research is an essential part of the whole work.

     In my opinion, after a few amendments indicated above, the manuscript could be forwarded to further procedure.

Round 2

Reviewer 1 Report

The manuscript improved significantly. I think the manuscript can be accepted in the current form.